# Computational epitope mapping of class I fusion proteins using low complexity supervised learning methods

**Marion F. S. Fischer**[1,2], **James E. Crowe**[3,4,5], **Jens Meiler**[1,2,6]*

**1** Department of Chemistry, Vanderbilt University, Nashville, Tennessee, United States of America, **2** Center for Structural Biology, Vanderbilt University, Nashville, Tennessee, United States of America, **3** Vanderbilt Vaccine Center, Vanderbilt University Medical Center, Nashville, Tennessee, United States of America, **4** Department of Pediatrics, Vanderbilt University Medical Center, Nashville, Tennessee, United States of America, **5** Department of Pathology, Microbiology and Immunology, Vanderbilt University Medical Center, Nashville, Tennessee, United States of America, **6** Institute for Drug Discovery, Leipzig University Medical School, Leipzig, Saxony, Germany

* jens@meilerlab.org

**Data Availability Statement:** All relevant data are within the manuscript and its Supporting information files. Code is available at the public repository https://github.com/mfsfischer/AxIEM.

## Abstract

Antibody epitope mapping of viral proteins plays a vital role in understanding immune system mechanisms of protection. In the case of class I viral fusion proteins, recent advances in cryo-electron microscopy and protein stabilization techniques have highlighted the importance of cryptic or 'alternative' conformations that expose epitopes targeted by potent neutralizing antibodies. Thorough epitope mapping of such metastable conformations is difficult but is critical for understanding sites of vulnerability in class I fusion proteins that occur as transient conformational states during viral attachment and fusion. We introduce a novel method Accelerated class I fusion protein Epitope Mapping (AxIEM) that accounts for fusion protein flexibility to improve out-of-sample prediction of discontinuous antibody epitopes. Harnessing data from previous experimental epitope mapping efforts of several class I fusion proteins, we demonstrate that accuracy of epitope prediction depends on residue environment and allows for the prediction of conformation-dependent antibody target residues. We also show that AxIEM can identify common epitopes and provide structural insights for the development and rational design of vaccines.

## Author summary

Efficient determination of neutralizing epitopes of viral fusion proteins is paramount in the development of antibody-based therapeutics against rapidly evolving or undercharacterized viral pathogens. Advances in the determination of viral fusion proteins in multiple conformations with 'cryptic epitopes' during attachment and fusion has highlighted the importance of epitope accessibility due to viral fusion protein flexibility, a physical trait not accounted for in previous B-cell epitope prediction methods. To consider how protein flexibility might influence antigenicity, viral fusion proteins must have been determined in conformations that correspond with multiple stages of attachment and/or fusion–and

**Funding:** All work was supported by the NIH funded 5U19AI117905-05 grant. The funders had no role in study design, data collection and analysis, decision to publish, or preparation of the manuscript.

**Competing interests:** The authors have declared that no competing interests exist.

have been extensively subjected to B-cell epitope mapping techniques. Despite advances in cryptic epitope determination, the available data is limited to a subset of class I fusion proteins that meet the above criteria. This poses a challenge to computational epitope mapping in generating an informative model that avoids overfitting. Here, we discuss a limited set of descriptive features, that when used in a variety of low complexity classifier models, matches or outperforms other publicly available B-cell epitope prediction methods in out-of-sample tests. From the models we tested, we use the linear regression model to highlight structural insights of epitopes and to demonstrate how this model may provide a novel approach to assess structural changes of antigenicity between viral fusion protein homologues.

## Introduction

Successful structure-based vaccine design relies on the identification of antigenic determinants that are most likely to elicit a humoral immune response, which can be achieved through the process of epitope mapping [1]. Given the time and cost of experimental methods used for epitope mapping, computational B-cell epitope prediction may provide a more practical starting point to narrow the search for commonly conserved or novel epitope targets. B-cell epitopes are defined as a spatially clustered set of antigen residues with a surface area of 600 $Å^2$ to 1,000 $Å^2$ that interact with an average antibody binding footprint of 800 $Å^2$ [2]. Epitope mapping of viral fusion proteins, however, indicates that multiple epitopes may overlap in a single domain or interface region, known as a site of vulnerability, but differ in binding site area and the combination of antigen residues that comprise a specific antigen-antibody interaction [3–8]. The major challenge of B-cell epitope mapping prediction is to identify residues that are most likely to form a direct contact with at least one neutralizing or protective antibody without assuming the conformation of the paratope. In addition, especially for emerging or under characterized antigens, the goal of computational epitope mapping is to not only have the sensitivity to detect previously determined sites of vulnerability, but also to provide reasonable predictions of novel sites of vulnerability.

Data collections for structural epitopes, especially residue-specific data, have increased in the past several years through the curation of databases such as the Immune Epitope Data Base (IEDB, iedb.org) [9] or the HIV Molecular Immunology Database (http://www.hiv.lanl.gov/content/immunology). This availability of data has permitted more accurate epitope prediction models, such as those provided by publicly available Discotope 2.0 or Ellipro discontinuous epitope prediction servers [10,11]. Even so, these epitope prediction models are limited to predicting epitopes of a single protein structure or even a single protein chain, which hinders the prediction of quaternary B-cell epitopes. Moreover, proteins are dynamic, *i.e.*, they assume more than one conformation, and a single conformation of a protein may not be sufficient to predict all possible epitopes. For instance, the stabilization of the respiratory syncytial virus (RSV) fusion (F) protein in its meta-stable prefusion conformation coincided with the identification of a broadly neutralizing epitope designated Site Ø, which is surface-inaccessible in the more stable postfusion F protein conformation [12].

The innate flexibility of class I viral fusion glycoproteins facilitates the entropy-driven process of membrane fusion to achieve cellular entry and host infection despite distinct fusion mechanisms. Compared to other proteins with antigenic determinants within a viral quasispecies, fusion proteins are more frequently targeted by broadly neutralizing antibodies, and therefore are prime candidates for rational structure-based viral vaccine design (so-called

'reverse vaccinology') [13–18]. As most viral fusion proteins are oligomeric and flexible, computational B-cell epitope prediction for these targets faces unique challenges. For thorough epitope mapping and prediction, the model should account for not only the prefusion quaternary structure of the target antigen, but also the changes in quaternary structure during attachment and fusion. Recent advances in experimental design and cryogenic electron microscopy (cryo-EM) allow discovery of cryptic epitopes in 'alternative' conformations of viral fusion proteins [19–25]. It is now feasible to identify residue-specific epitope accessibility changes during the fusion process, albeit with great effort for each antibody-antigen interaction.

We developed a machine learning approach designated Accelerated class I fusion protein Epitope Mapping (AxIEM) that harnesses evolutionary and structural features to classify whether a residue will reside within an epitope depending on the conformation of the fusion protein. We applied AxIEM to class I viral fusion proteins for which structures have been determined in at least two conformations and have been extensively subjected to experimental epitope mapping techniques. We show that AxIEM enables a higher out-of-sample success rate in defining viral fusion epitopes than previous methods and provides a computational tool to identify antigenic determinants of novel or under characterized viral fusion proteins.

## Results

### Description of AxIEM dataset

**Selection of protein structures.** The dataset used to build the AxIEM model includes six structure-based features that were obtained from a total of 34 PDB structures with 82 unique annotated epitopes. (S1 Table) The structures were selected based on the following criteria so that each protein ensemble included: i) at least two multimeric conformations of greater than 2.00 Å root mean square deviation (RMSD), ii) no two conformations of less than 1.00 Å RMSD, iii) every PDB had a resolution of below 4.5 Å, and iv) each conformation, such as an 'open' or 'closed' conformation, were equally represented within a protein ensemble [20,25–44]. The first criterion was imposed to maximize the structural diversity of each fusion protein ensemble, given that each protein includes at least one side chain contact rearrangement of over 10 Å between the precursor or prefusion conformation and the postfusion conformation [45–50]. Due to the limited number of fusion proteins that have been determined in both their prefusion and postfusion conformations, the threshold of 2.00 Å RMSD was used to allow for selection of multiple prefusion conformations of a single fusion protein, with a unique conformation defined by the second criterion. The last two criteria were needed since the feature set used to train AxIEM require structures with a resolution suitable for identifying side chain orientation without overrepresenting any single conformation's side chain contact orientations within an individual fusion protein ensemble.

**Feature selection and engineering.** For all 46,710 residues within the dataset, each residue possessed an expected classifier label that indicates the residue as an experimentally determined residue to be part of an epitope or not, plus a feature set of six metric values that account for the conformation and estimated energetic changes each residue undergoes during various stages of attachment and fusion (Fig 1). Three features were calculated to describe surface exposure, stability, and flexibility for each residue. (Fig 1) Relative solvent accessible surface area (SASA) exposure was calculated using the Neighbor Vector (*NV*) metric, which has been shown to accurately approximate SASA while being much less computationally expensive to compute than explicit SASA scores [44]. Additionally, *NV* is normalized within bounds of [0,1], so that SASA can be compared across protein species. The Rosetta-based per-residue total energy score (*REU*) was used to compute the single-body and pairwise interactions that a residue contributes to the energetic stability of a single conformation [45]. Lastly, contact

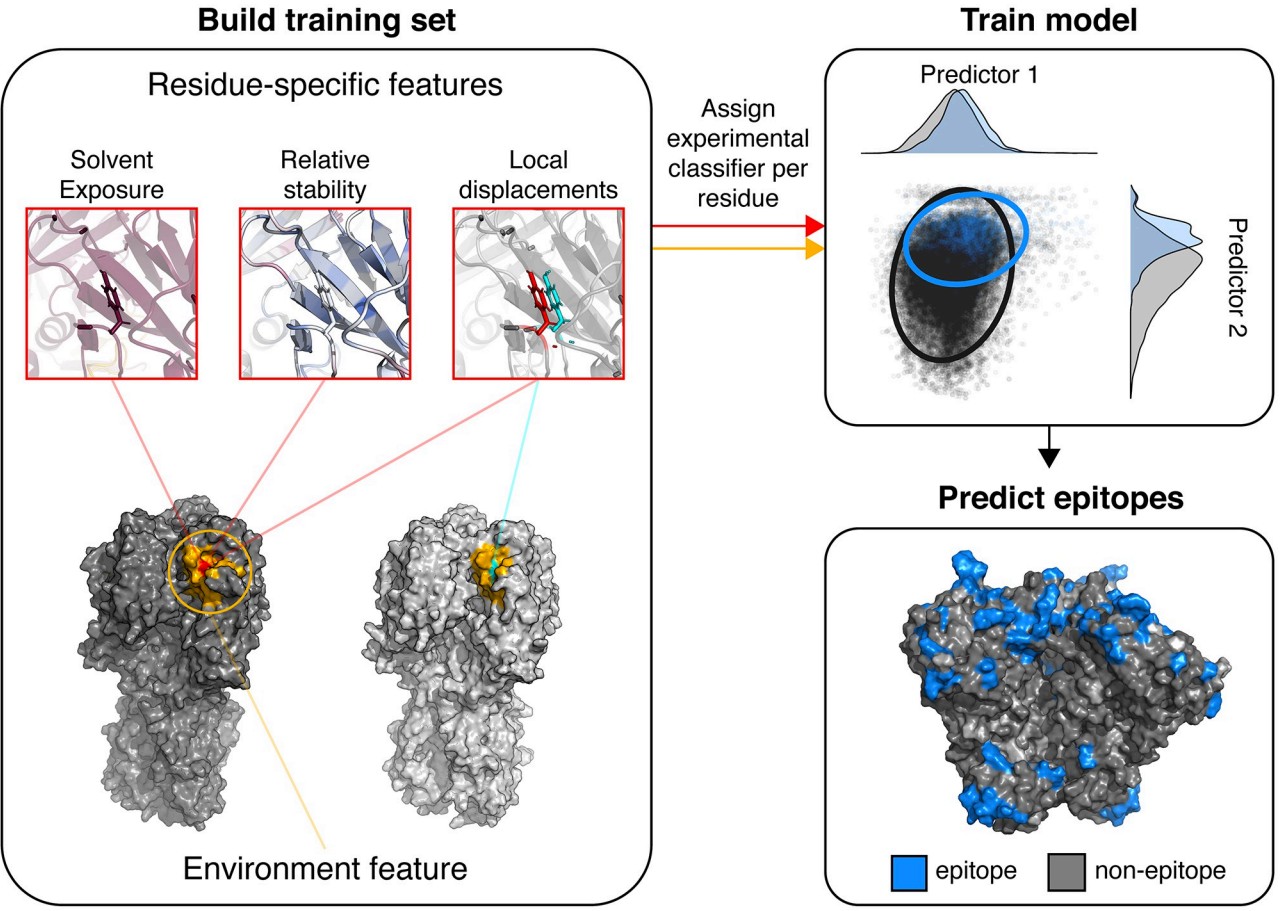

**Fig 1. Overview of AxIEM.** For each residue within the dataset, a set of six features was calculated that included three residue-specific features (outlined in red) and three neighbor features, with one for each residue-specific feature (outlined in dark yellow). Two of the residue-specific features are unique to each residue as a part of a single protein conformation that measured relative solvent exposure (as Neighbor Vector) and stability (as Rosetta Energy Unit). The feature measuring local displacements (as Contact Proximity variation) quantifies a residue's contact changes within an ensemble and relies on at least two conformations of aligned sequence for calculation. The environment features (as Neighbor Sums) approximate the relative solvent exposure, stability, and protein flexibility of the area surrounding the residue of interest. In addition to the six features, a classifier label is assigned to each residue. For training models, all related protein ensembles are removed from the dataset and withheld for testing. For simplicity, kernel density estimates of two features are depicted. Afterwards, the trained AxIEM model is used to predict classification of the left-out protein ensemble. The bottom right panel depicts AxIEM positive predictions for the HIV-1 Env trimer PDB ID 6CM3.

proximity variation ($CP_{var}$) measures the estimated distance changes of the chain contacts a single residue undergoes within a protein ensemble as estimated by Cβ atom orientation variation to all other Cβ atoms [51].

Comparison by a Welch's two-tailed t-test for each residue-specific feature indicated that the mean value differed significantly between residues that have and have not been experimentally determined to form an antibody (Ab) binding interaction, with $p < 1.00 \times 10^{-30}$ for all three features (S1 Fig). We also calculated a distribution-free overlap score η to estimate the similarity of score distributions for epitope and non-epitope residues, with an η of zero indicating the two distributions are unique and an η of one indicating identical score distributions [52]. The overlap of residue-specific scores was relatively high, with η = 0.610 for *REU* values, η = 0.738 for $CP_{var}$ values, and η = 0.791 for *NV* values.

To see if we could engineer features that are more descriptive of epitope and non-epitope residues, we created the Neighbor Sum (*NS)* metric to estimate the local environment

surrounding a single residue as a cosine-weighted sum for each $NV$, $REU$, and $CP_{var}$ feature, denoted as $NS_{NV}$, $NS_{REU}$, and $NS_{CP}$ respectively. The motivation for creating each $NS$ score term was to examine whether a residue is more likely to form an epitope if its surrounding residues exhibit similar $NS$ feature scores. To avoid making assumptions about what distance away neighboring residues contribute to the antigenicity of a residue, we used an upper boundary cutoff of incremental distances of 8 Å, or half of the radius of an average Ab binding footprint, up to 64 Å to determine which optimal neighborhood volume to use to compute NS. We used a lower boundary cutoff of 4.0 Å to limit the weighted contribution of 1 to only the residue itself, not its neighboring residues. (S2 and S3 Figs) $NS_{REU}$ values indicated that epitope residues are much more likely to reside in energetically frustrated regions (*i.e.* higher energy regions), with the separation in distribution overlap $\eta \leq 0.579$ and significance in separation of mean $NS_{REU}$ values of $p \leq 2.31 \times 10^{-241}$ for all upper boundary cutoff values. The $NS_{NV}$ metric revealed that although epitope residues have higher relative surface exposure than non-epitope residues, epitope neighbors are much more likely to be shielded from the protein surface, with $\eta \leq 0.807$ and significance in separation of mean $NS_{NV}$ values of $p \leq 2.31 \times 10^{-241}$ for all upper boundary cutoff values. Differences in $NS_{CP}$ distributions between epitope and non-epitope residues were inconsistent in overlap and significance for each upper boundary radius used to calculate $NS_{CP}$, suggesting that a residue's own flexibility contributes to the antigenicity of a residue but not the flexibility of its neighboring regions. Please refer to Methods and S1 Protocol Capture for the detailed description of each metric and definition of epitope classifier labels.

**Definition of performance accuracy.** Performance accuracy relied on the definition of a true positive ($TP$) as a residue with a prediction score above a certain prediction score threshold value and was designated as an expected epitope residue with a classifier label of '1' prior to building the model. A true negative ($TN$) is a residue with a prediction score below the same prediction score threshold value and was designated as an expected non-epitope residue with a classifier label of '0'. It should be noted that this definition is not as rigid as a $TP$ given the possibility of unidentified or incomplete characterization of antigenic sites. A false negative ($FN$) or false positive ($FP$) is any residue that scores incorrectly below or above, respectively, of the same given threshold.

## AxIEM more consistently improves epitope mapping prediction accuracy

We trained and tested four low-complexity supervised learning methods, including linear regression, Bayes classification, logistic regression, and random forest classification to avoid overfitting the limited dataset. Construction of a training set involved withholding all feature data originating from any PDB structures representing one of the seven class I fusion proteins. The withheld feature data were retained as the corresponding leave-out test set. When either influenza H3 and H7 hemagglutinin or Severe Acute Respiratory Syndrome (SARS)-related Spike (S) proteins, SARS-CoV S and SARS-CoV-2 S, feature data were used as the test set, the related hemagglutinin or S protein feature data were withheld from the training set but were not included within the test set to ensure non redundancy of feature data during out-of-sample performance evaluation. Please refer to Methods for parameterization details used for each statistical model.

We found that the mean AUC scores peaked using an $NS$ upper boundary radius of 24 Å, with AUC values of 0.771±0.0475, 0.759±0.0472, and 0.695±0.0730 for linear regression, Bayes classification, and random forest classification, respectively using all three $NS$ features. (S4 and S5 Figs) We also performed leave-out using feature sets that excluded one of the three $NS$-based features from the training and test sets. Regardless of which $NS$ feature was excluded,

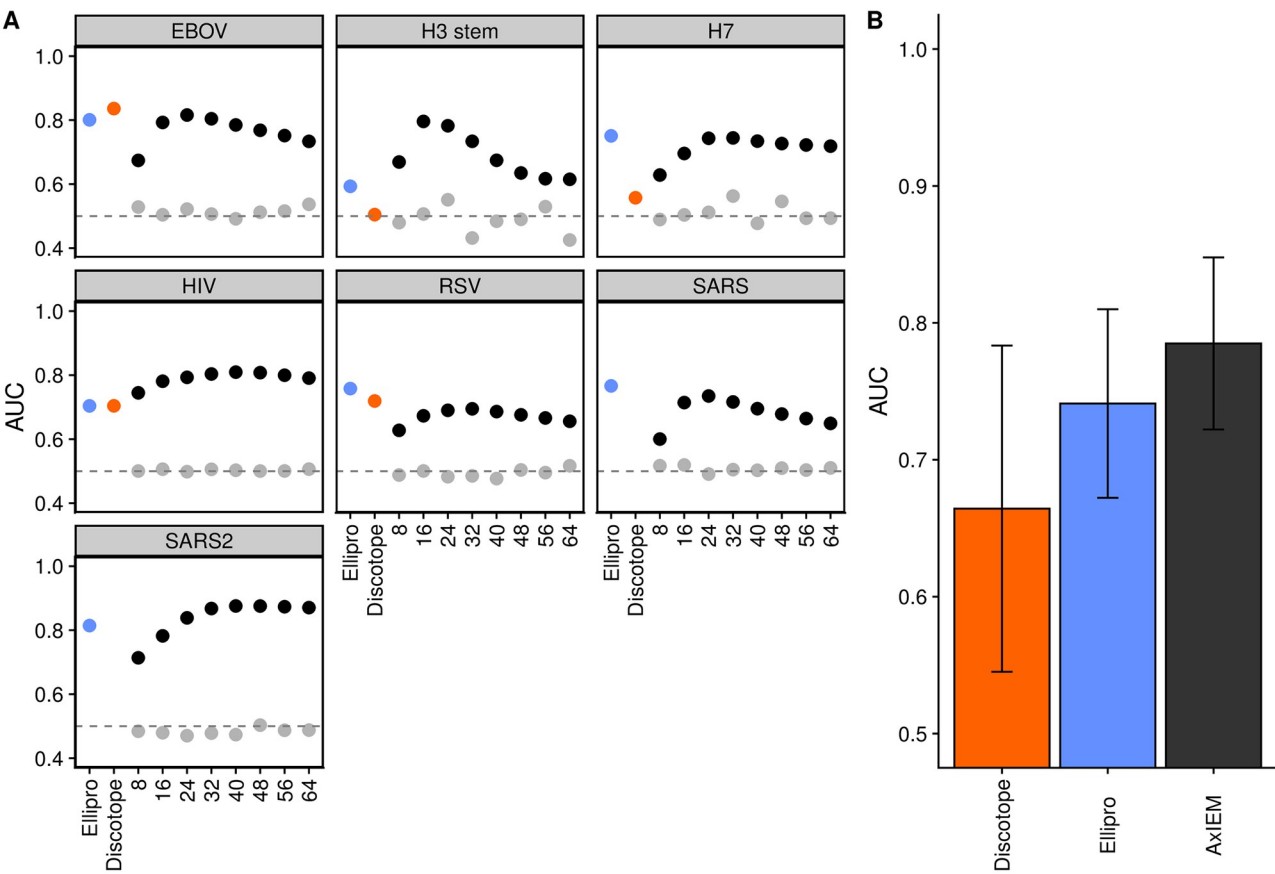

**Fig 2. Performance comparison of discontinuous epitope prediction methods with AxIEM.** (**A**) Comparison of AUC values by virus using the highest performing AxIEM model. Each panel represents the determined AUC value for an individual test set where the feature sets of all PDBs for the fusion protein listed as the panel title were withheld from the training data and used as the test dataset. In the case of influenza and SARS-related proteins, all related structure data were withheld from the training set. Black dots indicate that the feature set {$NV$, $REU$, $CP_{var}$, $NS_{NV}$, $NS_{REU}$} was used to train a linear regression model, with the upper boundary radius used to calculate $NS_{NV}$ and $NS_{REU}$ listed along the x axis as integers. Light grey represents a negative control for which each residue was assigned a random value from a normal Gaussian distribution as determined by the mean and standard deviation of each feature's original values. The randomized feature set is expected to have an AUC of 0.5 to indicate that the performance of unrandomized training data was not by chance. Note that Discotope was not able to evaluate predictions for SARS-CoV and SARS-CoV-2 S proteins due to protein size and server time limits. (**B**) Comparison of average AUC values. Bar height indicates the average AUC value for individual test sets for each method, with the two-tailed sample standard deviation indicated by the line heights.

mean AUC scores also peaked at 24 Å, with the top performing method being linear regression using the feature set {$NV$, $REU$, $CP_{var}$, and $NS_{REU}$} with a mean AUC score of 0.785 ±0.0628. (Fig 2) When all NS features were excluded, mean AUC performance dropped to 0.620±0.0626 using linear regression, and the largest drop in mean AUC performance using $NS$ features was when $NS_{NV}$ feature was excluded, with a value of 0.714±0.0714 using a 24 Å radius to calculate the $NS$ features. (S5 Fig) For a performance comparison, we computed the average AUC performance of Discotope 2.0 and Ellipro epitope prediction servers available through the IEDB using default parameters and the same residues used to perform the AxIEM withheld test sets, which we found to be 0.664±0.119 and 0.741±0.0689 respectively.

## AxIEM is a tool to identify common sites of vulnerability and their conformations

Epitope mapping of B-cell epitopes can be achieved either explicitly through structural determination of an Ab-antigen complex or implicitly through peptide-based, mutagenesis, or

hydrogen/deuterium exchange methods that require a reference structure to obtain the conformation of the site of vulnerability [53–56]. Using a reference structure to map out overlapping epitopes requires either prior knowledge of the conformation to which an epitope is accessible, such as RSV F Site Ø, or assumes that the representative antigen structure resembles the major subpopulation of all possible antigen structures [57,58].

SARS-CoV and SARS-CoV-2 S proteins provide a unique challenge for structural epitope mapping because multiple conformations of each trimeric S protein have been determined with various orientations of the receptor binding domain (RBD), and with each conformation associated with unique mechanisms of neutralization [59–61]. Structurally determined conformations includes the 'closed', '1-RBD open', and '2-RBD open' conformations for both S proteins [41,44,59–61], and the '3-open'/'fully open' conformation for SARS-CoV-2 S protein [40]. The majority of available structures of Ab-S protein interactions, however, exist as fragment Ab and RBD complexes and are not annotated as specific to any one or more conformations [20]. To map the fragment Ab-RBD epitopes to a reference S protein conformation requires superimposition of the RBD from the fragment Ab-RBD interaction to each of the RBDs within each conformation to determine if the epitope is accessible, both in terms of epitope surface accessibility and potential Ab clashes with other protomers. Given the criteria used by this study to map epitopes to each conformation, most fragment Ab-S protein complex epitopes were exclusively mapped to the closed conformation for both SARS-CoV and SARS-CoV-2 S proteins, which were also used to define the expected classification label of whether a residue should be predicted to be an epitope or not. AxIEM predictions conversely suggest that the open RBD conformations are more likely to be antigenic. (Fig 3A) These predictions are corroborated by recent structural studies that were performed after the dataset used to train the AxIEM model was curated and depict broadly neutralizing mechanisms that exclusively target open conformations through recognition of quarternary epitopes [59,60,62,63]. Using the same epitope definition as AxIEM, Ellipro outperformed AxIEM when predicting SARS-CoV S epitopes, with an AUC of 0.766 compared to 0.735. However, Ellipro failed to predict any epitopes for the '1-open' and '2-open' conformations of SARS-CoV S protein, whereas AxIEM predicted novel conformation-dependent B-cell epitopes within these open conformations which were later identified as antigenic targets of broadly neutralizing Abs. (Fig 3A)

Another critical aspect of epitope prediction with regards to fusion proteins is being able to identify common sites of vulnerability and to understand the structural differences between them in related proteins to develop immunogens for next-generation vaccines. We compared the structural and sequence similarity of AxIEM predicted RBD epitopes for SARS-CoV and SARS-CoV-2 RBD specific to the '2-open-RBD' conformations and found 55% sequence conservation, and a 19.4 Å RMSD structural difference between epitopes (Fig 3B and S2 Table). Likewise, we aligned SARS-CoV and SARS-CoV-2 predicted NTD epitopes and found no sequence or structural similarity. Given these data, identification of a broadly neutralizing Ab against multiple SARS-related S proteins is constrained not only by sequence similarity, but also conformation similarity and availability as metastable up or open conformations.

## Discussion

### AxIEM provides a low complexity solution to epitope mapping

For this study, we chose a final model that employs linear regression, in conjunction with a limited feature set to avoid overfitting from the experimentally validated dataset. Despite its low complexity, the AxIEM model improves prediction of tertiary and quaternary epitopes of class I viral fusion proteins compared to the IEDB sponsored discontinuous epitope prediction

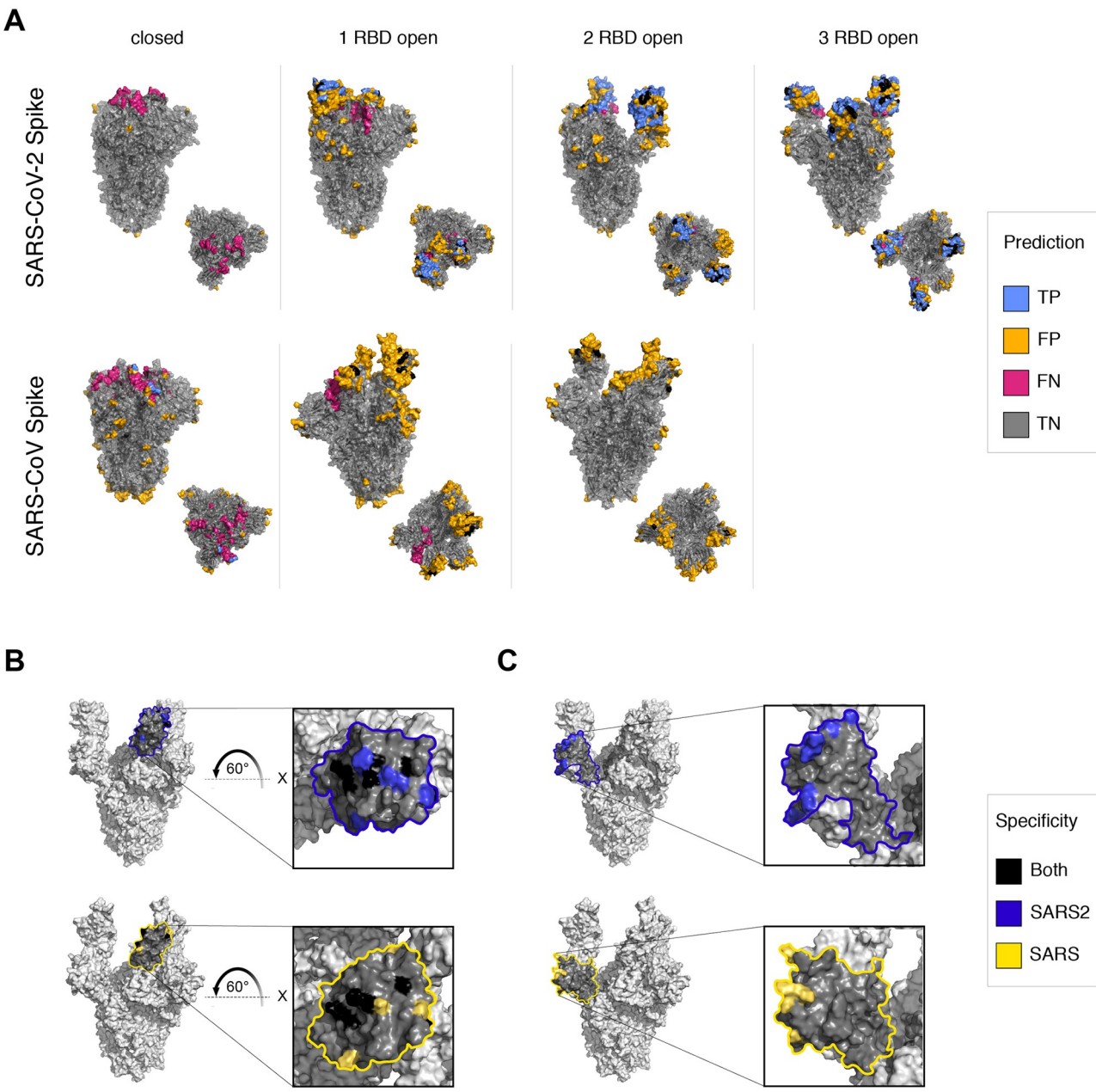

**Fig 3. Overview of AxIEM predictions of coronavirus Spike protein epitopes.** (**A**) Predictions mapped to conformation models. Side and top views are shown for each protein and conformation. Black indicates alignment of positive predictions. (**B**) Alignment of common RBD epitopes. Highlighted area represents all residues that are within 16 Å of the geometric centroid of identified common epitopes. The models used to represent the SARS-CoV-2 (top) or SARS-CoV (bottom) include PDB models 7CAK and 6NB7. Black indicates alignment of positive predictions sharing the same sequence identity. Blue and yellow indicate sequence position alignment only of positive predictions. (**C**) Alignment of novel NTD epitopes. Highlighted areas, model representation, and coloring are the same as in panel B.

methods, Ellipro and Discotope 2.0. The limited computational requirements of AxIEM, either to use as is or retrain, provides an accessible tool for vaccine development strategies such as screening for novel or cryptic antigenic sites of newly determined class I viral fusion proteins, comparing fusion protein homologues as demonstrated in Fig 3, or employing AxIEM within subunit vaccine design platforms. The further use of AxIEM as a computational epitope

mapping strategy, however, requires further consideration of aspects of viral structural biology and poses future challenges to generalize epitope prediction, as discussed in the following sections.

## Blocking the moving target requires dynamic precision

To model viral fusion protein flexibility, AxIEM requires at least two conformations to represent major conformation populations during fusion protein-mediated cellular entry and relies on the coarse-grained flexibility metric $CP_{var}$ to estimate cumulative local residue contact displacement. Almost certainly, two conformations are insufficient to fully summarize the biophysical changes of viral entry in terms of representing the major subpopulations, dynamics, pH, *etc.* when entering the host cell. Exclusion of $NS_{CP}$ from the AxIEM feature set insignificantly affected overall performance. (S4 Fig) However, AxIEM exceeds in classifying protein antigenic residues when prefusion conformation heterogeneity is more thoroughly represented, such as in the case of ebola glycoprotein, HIV Env, or SARS-CoV-related S proteins that have multiple prefusion conformations included in the datasets. To improve accurate calculation of $CP_{var}$, one could use recently developed machine learning based structure predictions methods like AlphaFold to predict representative structures of alternative conformations of fusion proteins using an approach described by del Alamo and colleages [64,65].

Conversely, the AxIEM model overestimates the probability of antigenicity of postfusion and membrane-proximal external region (MPER) protein surfaces regardless of conformation, as displayed S6 –S10 Figs possibly because these regions are not truly surface-accessible to an Ab in a cellular environment, which likely explains the poor individual performance of predicting RSV and influenza H3 stem epitopes due to the large fraction of surface-accessible residues within each postfusion conformation. AxIEM was also more likely to predict epitopes proximal to membrane regions. In the case of HIV Env MPER, antibodies like 10E8 require HIV Env to 'tilt' in relation to the lipid bilayer to gain surface accessibility [66], and it is possible that similar regions may present true sites of vulnerability. Further validation would require either identification of novel neutralizing antibodies like 10E8 or better quantification of membrane or protein crowding. Additionally, any information regarding glycosylation patterns was not included in the AxIEM model due to the lack of high resolution (< 3.0 Å) determination of most glycosylation sites, and therefore any predictions made by AxIEM would need to be supplemented with glycosylation modeling to further assess validity of any initial AxIEM predictions. Overall, AxIEM's performance and other computational epitope prediction methods would likely benefit from modeling of protein target dynamics and major subpopulation states to better interrogate how protein flexibility affects antigenicity and which major subpopulation states are most likely to elicit a strong neutralizing response.

## Methods

### Explanation of epitope predictors

*Neighbor Vector* (*NV*) The per-residue solvent-accessible surface area (SASA) metric *NV* approximates the proximity and spatial orientation of surrounding residues to estimate relative surface exposure of a residue, as previously described [4] In brief, *NV* employs the Contact Proximity (*CP*) and Neighbor Count (*NC*) algorithms to calculate the sum of each surrounding residue's $C_\beta$-$C_\beta$ distance (*d*) unit vector weighted by its likelihood to make a contact with the reference residue, calculated as *CP* (Eq 1), and is normalized by the sum of all possible contacts within the residue's vicinity, calculated as *NC* (Eq 2). In other words, for a highly buried residue, the weighted *d* unit vectors of all $C_\beta$-$C_\beta$ distances will be directed outwards in many directions so that its *NV* score $\approx \cong 0$, whereas a highly exposed residue's weighted *d* unit

vectors will be directed uniformly so that its $NV$ score $\cong 1$. We use the lower and upper boundaries of 4.00 Å and 12.8 Å, respectively, because 4.00 Å was shown to be the optimal lower boundary to accurately assess per-residue SASA using $NC$ and 12.8 Å is the maximum length of a $C_\beta$-$C_\beta$ distance where any atom of one amino acid's side chain has been shown to make a direct interaction with another amino acid's side chain (Eq 3).

$$CP = \begin{cases} 1, & if\ d \leq 4.00\ \text{Å} \\ 0, & if\ d \geq 12.8\ \text{Å} \\ \frac{1}{2}\left[\cos\left(\frac{d - 4.00\ \text{Å}}{12.8\ \text{Å} - 4.00\ \text{Å}} \times \pi\right) + 1\right], & if\ 4.00\ \text{Å} < d < 12.8\ \text{Å} \end{cases} \tag{1}$$

Where $d$ is the geometric distance between two $C_\beta$ atoms of two residues. In the case of glycine, a dummy $C_\beta$ atom was used in place of its hydrogen.

$$NC_i = \sum CP_j(d(i,j), lower,\ upper) \tag{2}$$

Where $CP_j$ is the evaluated $CP$ score of the $j$th residue in relation to the residue of interest $i$ given the lower boundary of 4.00 Å and the upper boundary of 12.8 Å.

$$NV_i = \left\|\frac{\sum_{i=1}^{j}\left(\vec{v}_{ij}/\|\vec{v}_{ji}\| \times CP_j\right)}{NC_i}\right\| \tag{3}$$

where $\vec{v}_{ij}/\|\vec{v}_{ji}\|$ is the unit vector of the $j$th residue multiplied by the $CP$ score of the $j$th residue, both terms in relation to the $i$th residue.

*Per-residue Rosetta Energy Unit (REU)* The approximation of the relative Gibbs free energy for each residue of a minimized single protein conformation was calculated with the Rosetta *ref2015* energy function and using the jd2_scoring application to estimate the single body and pairwise interaction energies of a residue as the Rosetta per-residue total energy score, which is reported in Rosetta Energy Units (*REU*).

*Contact Proximity variation (CP$_{var}$)* The metric $CP_{var}$ has previously been used to estimate the relative local side chain contact changes of a single residue experiences as part of a protein ensemble and was calculated as the sample variance of likely contacts a single residue will form (Eq 4) [67]. This was estimated by $CP$ for each $C_\beta$-$C_\beta$ distance, again using 4.00 Å and 12.8 Å as the lower and upper boundaries.

$$CP_{var}(i) = \frac{1}{n-1}\sum_{k=1}^{n}\sum_{i=1}^{j} CP_{ijk}^2 - \overline{CP_{ij}^2} \tag{4}$$

where $i$ is the residue of interest, $j$ is another residue in the same protein conformation, and $n$ is the number of protein conformations within the protein ensemble.

*Neighbor Sum (NS)* The $NS$ metric was calculated as a cosine weighted sum of either $NV$, $REU$, or $CP_{var}$ (Eq 5). In the Results section, we reported $NS$ using a fixed lower boundary of 4.00 Å so that only residue $i$ contributes with a weight of 1 and adjusted the upper boundary to test for epitope radius size as noted (Eq 6).

$$NS_u(i) = \sum_{i=1}^{j} w_{ij} \times feature(i) \tag{5}$$

where $i$ is the residue of interest, $j$ is another residue in the same protein conformation, $u$ is the upper boundary radius at which surrounding residues contribute to $i$ residue's $NS$ value, and $j$

residue's weighted contribution $w$ to residue $i$ is determined in Eq 6.

$$w = \begin{cases} 1, & \textit{if } d \leq 4.00\,\text{Å} \\ 0, & \textit{if } d \geq upper \\ \dfrac{1}{2}\left[\cos\left(\dfrac{d - 4.00\,\text{Å}}{upper - 4.00\,\text{Å}} \times \pi\right) + 1\right], & \textit{if } 4.00\,\text{Å} < d < upper \end{cases} \tag{6}$$

## Definition of conformation-specific epitopes

All epitope residues were first identified as any residue that has been annotated as an epitope by the IEDB, Influenza Research Database's Immune epitope search, or the HIV Molecular Immunology Database that is associated with a PDB structure. IEDB searches used the filters 'Positive Assays only', 'Epitope Structure: Discontinuous', 'No T cell assays', 'No MHC ligand assays' and 'Host: Homo sapiens' as of June 1, 2020. Influenza epitope searches used the filters 'Virus Type A', 'Subtype H3 (or H7)', 'Protein HA, Segment 4', 'Experimentally Determined Epitopes', 'AssayType Category and Result B-cell Positive', and 'Host Human' as of June 1, 2020. HIV-1 epitopes include human epitopes as listed in the interactive epitope maps as of June 1, 2020.

To determine which each epitope residue's conformation specificity, a residue must have at least one PDB structure of an Ab-antigen complex where it has been annotated as an epitope structure (*i.e.* it has an IEDB/other identification number) and that the PDB associated with the epitope, when aligned to a protomer, results in no atomic overlap of the antibody with the whole PDB structure. Checking for overlap was performed as follows: i) For each PDB antibody-antigen structure annotated to be associated with a unique epitope ID, the antibody and antigen were created as independent PyMOL objects, for example 'antibody+antigen' and 'antigen'. ii) Three PyMOL objects were created for each AxIEM benchmark PDB of identical species to the 'antigen' object, labeled as 'objA', 'objB', 'objC', respectively, for each protomer. iii) The 'antigen' object was first aligned to 'objA'. iv) Next the antigen of the 'antibody+antigen' object was aligned to the 'antigen' object to check for van der Waals overlap of the Ab with the protomer. The residues annotated within a single epitope ID were considered to associated, or mapped, to that protomer only if zero atom coordinates of the Ab structure within the 'antibody+antigen' object were within 0.6 Å of any atoms within the PDB containing the protomer of interest. Steps iii and iv were repeated for 'objB' and 'objC'.

Note that it was possible for one epitope ID and associated PDB to fail the overlap test for a single protomer, while another epitope ID containing similar or identical residues passes the overlap test with an alternative Ab-antigen complex representation. In this case, if any protomer residue passed the overlap test for at least one annotated Ab-antigen complex as described in step iv, that residue was labeled as an epitope residue, or '1' within the feature dataset, *e.g.*, AxIEM.data or AxIEM_*.data. The majority of residues classified as an epitope according to our definition, listed in S1 Protocol Capture, were represented in multiple Ab-antigen complexes–more than 80% of epitope residues have been found to form an epitope interface with at least two unique Abs and more than 75% of epitope residues have been determined in at least two structural representations of the same Ab-antigen complex. Therefore, the structural diversity represented within the AxIEM epitope classifier labels ensures that possible side chain rearrangements due to Brownian motion are represented, despite the stringent 0.6 Å cutoff value to detect van der Waals overlap of an Ab atom with an atom within the fusion protein conformation of interest, while also accounting for Ab binding angle that might prohibit binding to that protomer given the backbone conformation of the trimeric fusion protein.

The AxIEM_updated.data file includes all feature set values, classifier labels, and associated viral fusion protein, PDB ID, and PDB residue ID labels that were used for the evaluation of AxIEM.

## Structure preparation

Curation of viral fusion protein conformations began with a Protein Data Bank query of viral fusion proteins in their trimeric state that were identical in virus family and strain or serotype. All PDB structures that shared $\geq$ 95% amino acid sequence identify were downloaded from the Protein Data Bank and were superimposed in a pairwise fashion using the PyMOL align command with number of cycles set to zero. For PDB structures sharing less than 1.0 Å RMSD, the structure with the least number of missing densities was selected to represent a single fusion protein conformation. The final set of selected fusion protein ensembles included ensembles with at least a single 2.0 Å pairwise RMSD between two conformations, and no two conformations shared less than 1.0 Å RMSD. Additionally, for ensembles such as SARS-CoV-2 that have multiple, distinct conformations the number of distinct conformations were equally balanced in conformation representation.

Next, because the feature $CP_{var}$ requires that all PBD structures within an ensemble share 100% sequence alignment (but not sequence identity), PDB fasta files were aligned using Clustal Omega (www.ebi.ac.uk/Tools/msa/clustalo/) to identify residues that were structurally resolved for all equivalent chains and in all conformations of each viral protein. Residues that were not determined as such were manually removed using PyMOL to generate an aligned pdb file. Since some structural models contained mutations for stabilization, the consensus sequence was used to replace the initial sequence. The consensus sequence was obtained by performing a multiple sequence alignment of all available full-length sequences with Clustal Omega and using the EMBOSS cons package (ftp://emboss.open-bio.org/pub/EMBOSS) to identify the consensus sequence. The Rosetta partial_thread application was used to thread, or replace, each residue with the corresponding consensus sequence. Afterwards, the threaded structures were subjected to a constrained relax using the Rosetta FastRelax application to generate 100 models for each protein. From the generated models, the model with the combined lowest energy and lowest structural RMSD to the respective aligned structure was selected as the final model used to evaluate $NV$, $REU$, $CP_{var}$, and $NS$-weighted features. For a complete guide to which residues and methods were used for model generation, please refer to S1 Protocol Capture.

$$rmsd_p = \frac{1}{n} \sqrt{\sum_{i=1}^{n} \left| \sum_{j=1}^{n} d_a(i,j) - d_b(i,j) \right|} \tag{7}$$

$$d = \left(x_j - x_i\right)^2 + \left(y_j - y_i\right)^2 + \left(z_j - z_i\right)^2 \tag{8}$$

## Statistical models

Linear regression, logistic regression, and random forest classification were implemented in Python using Scikit learn [68]. Models generated using linear regression and random forest used the default parameters. Transformation of the feature set to a multivariate Gaussian distribution and training of Bayes classifier models were implemented in Python using the pomegranate package [69]. Each of the four statistical methods used seven training sets that exclude all structural data of PDBs related to the withheld data for testing as previously described. Logistic regression models used the Scikit Learn MinMaxScaler to scale each feature set to bounds [0,1]; however, logistic regression models were not discussed in the Results section

because all models failed to converge, possibly due to the presence of local maxima in the log-likelihood estimates [70]. Statistical analysis for ROC curves and AUC values were performed using Python. Conductance of the Welch's two-tailed t-test $p$ value and calculation of the distribution-free overlapping index η were performed in R. All AUC scores reported represent out-of-sample performance. No retraining was performed. Mean AUC scores and their sample standard deviations reported in Fig 2B were calculated in Excel.

## Computational resources

All calculations were performed on a Core i9-9980HK laptop with 16 GB RAM and Gentoo Linux operating system. All datasets, code, and documentation used for this study are publicly available at https://github.com/mfsfischer/AxIEM.

## Supporting information

**S1 Fig. Distributions of residue-specific features of epitope from non-epitope residues.** Mean feature values of epitope and non-epitope residues are indicated by vertical dashed lines. Score values within one standard deviation of each group's mean are shaded and encased by vertical dotted lines. The distribution-free overlapping index η and the Welch's two-tailed t test p value are indicated in each panel for the distribution overlap and significance of mean difference between epitope and non-epitope residues. For reference, two distributions would be identical if η = 1.00, and unique if η = 0.00. A p value of less than 0.05 indicates significant differences in mean values. A) Neighbor Vector (NV) distributions. B) Per-residue Rosetta Relative Energy Unit (REU) distributions. C) Contact proximity variation (CP) distributions. (PNG)

**S2 Fig. Comparison of Neighbor Sum overlap by feature.** The upper boundary radius used to calculate each NS feature is indicated in each panel's title. The distribution-free overlapping index η and the Welch's two-tailed t test p value are indicated in each panel for the distribution overlap and significance of mean difference between epitope and non-epitope residues' NS values. Vertical dashed lines indicate each distribution's mean value. For values p*, the p value could not be calculated due to values being too close to zero. (PNG)

**S3 Fig. Comparison of Neighbor Sum overlap by feature (continued).** (PNG)

**S4 Fig. AUC performance comparison of models and feature sets.** (PNG)

**S5 Fig. Summary of AUC performance using alternative NS feature sets.** (PNG)

**S6 Fig. AxIEM predictions of ebola virus glycoprotein.** Proteins are oriented so that residues closest the viral membrane are at the bottom. Predictions are color coded using the same scheme as in Fig 3A, with TP as blue, FP as yellow, FN as pink, and TN as grey. (PNG)

**S7 Fig. AxIEM predictions of influenza hemagglutinin (H3) stem.** Proteins are oriented so that residues closest the viral membrane are at the bottom. Predictions are color coded using the same scheme as in Fig 3A, with TP as blue, FP as yellow, FN as pink, and TN as grey. (PNG)

**S8 Fig. AxIEM predictions of influenza hemagglutinin (H7).** Proteins are oriented so that residues closest the viral membrane are at the bottom. Predictions are color coded using the same scheme as in Fig 3A, with TP as blue, FP as yellow, FN as pink, and TN as grey.
(PNG)

**S9 Fig. AxIEM predictions of human immunodeficiency virus 1 envelope protein.** Proteins are oriented so that residues closest the viral membrane are at the bottom. Predictions are color coded using the same scheme as in Fig 3A, with TP as blue, FP as yellow, FN as pink, and TN as grey.
(PNG)

**S10 Fig. AxIEM predictions of respiratory syncytial virus fusion protein.** The RSV F structure 4MMS is oriented so that residues closest the viral prefusion membrane are at the bottom. The RSV F structure 3RKI is oriented so that residues closest the viral postfusion membrane are at the top. Predictions are color coded using the same scheme as in Fig 3A, with TP as blue, FP as yellow, FN as pink, and TN as grey.
(PNG)

**S1 Table. Protein Data Bank (PDB) accession identities and resolution of models used for the AxIEm dataset.**
(TIF)

**S2 Table. Residue identities of AxIEM predicted epitopes.** Clustal Omega was used to perform a multiple sequence alignment of SARS-CoV and SARS-CoV-2 Spike protein sequences. Aligned residues are indicated when residue identities are present in both the SARS-CoV and SARS-CoV-2 columns. Residues of identical sequence identity are indicated in bold. Residue numbers correspond to PDB ID 6NB7 (SARS-CoV S, 2-up conformation) and 7CAK (SARS-CoV-2, 3-up conformation), and chain B for both models. One-letter sequence identities correspond to the consensus sequence. Note, consensus sequence residue H432 of SARS-CoV S protein was altered from the original 6NB7 sequence Y432, which would be identical to aligned Y449 of SARS-CoV-2.
(TIF)

**S1 Protocol Capture. Protocol capture and data curation details.**
(PDF)

## Acknowledgments

We thank Dr. Axel Fischer for his helpful insights.

## Author Contributions

**Conceptualization:** Marion F. S. Fischer.

**Data curation:** Marion F. S. Fischer.

**Formal analysis:** Marion F. S. Fischer.

**Funding acquisition:** James E. Crowe, Jens Meiler.

**Methodology:** Marion F. S. Fischer.

**Software:** Marion F. S. Fischer.

**Writing – original draft:** Marion F. S. Fischer.

**Writing – review & editing:** James E. Crowe, Jens Meiler.

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
