## [Decision Letter · Decision Letter 0]

3 Jul 2022

Dear Dr. Fischer,

Thank you very much for submitting your manuscript "Computational epitope mapping of class I fusion proteins using Bayes classification" for consideration at PLOS Computational Biology.

As with all papers reviewed by the journal, your manuscript was reviewed by members of the editorial board and by several independent reviewers. In light of the reviews (below this email), we would like to invite the resubmission of a significantly-revised version that takes into account the reviewers' comments.

As you can see below, the reviewers found your work of interest. Please address their concerns regarding the datasets and github repository, as well as other minor comments.

We cannot make any decision about publication until we have seen the revised manuscript and your response to the reviewers' comments. Your revised manuscript is also likely to be sent to reviewers for further evaluation.

Sincerely,

Dina Schneidman

Software Editor

PLOS Computational Biology

Reviewer's Responses to Questions

**Comments to the Authors:**

Reviewer #1: Computational epitope mapping of class I fusion proteins using Bayes classification

Mapping of B-cell epitopes onto structure is a major experimental and computational challenge. Here, Fischer et al. propose a computational approach based on machine learning for prediction of B-cell epitopes from class I viral fusion proteins. Their approach, AxIEM, consists in calculating four residue-wise features, followed by a Bayesian classifier. They test it in a leave-one-out setting on seven viral fusion proteins, each featuring multiple conformations. They find that their method outperforms two widely used methods, Discotope and Ellipro, in terms of sensitivity/specificity. They predict epitopes for multiple conformations of the SARS-CoV-1/2 spike protein trimer, and investigate epitope conservation across conformations and sequences.

Overall, the proposed approach is well-founded and could indeed outperform baseline methods. However, the manuscript lacks clarity and key information was missing (e.g. definition and source of the epitope labels, rationale for choosing the structures out of many across the PDB). I also have concerns regarding the validity of the benchmark, as: i) the scale of the experimentation is limited and multiple fusion proteins were left out (e.g. hemagglutinin for other influenza strains which have distinct epitopes) and ii) inspection of the Github repository raised substantial concerns regarding the epitope data used, as it seems incomplete. Finally, the GitHub repository is difficult to use as it lacks any documentation, and there is no simple way to run the model on a different example. Consider providing a script taking as inputs a list of PDB files and outputting the residue-wise epitope probabilities.

Regarding the model:

1) Usage of Bayes classifier is rightfully motivated by the low sample size. However, how accurate is the multivariate gaussian distribution assumption for this data set? Especially given that the NV feature is 0-1 bounded with an heavy left tail, and that the CP RMSD feature is clearly bimodal for non-epitope residues (Sup. Fig1). Have other supervised ML approaches such as logistic regression or random forest been evaluated here?

2) Regarding the third feature. The calculation of the standard deviation in Eqn 4 critically depends on the conformational ensemble selected. For the SARS-CoV-2 spike protein, hundreds of cryo-EM structures are available, raising the question of how the 8 structures were selected. Were all the main conformational states represented (from fully closed to fully open)? If yes, how was the balancing done between the four main states? (0,1,2,3 open RBDs?).

3) L355: “In the case where only two conformations were used, only the mean CP value was calculated.” The variance is well-defined even for only two samples; why not use it? Also, for such small number of conformations, it is more reliable to use the unbiased estimate for the variance, by dividing by n-1 rather than n.

4) Eqn5 L363: Eqn5 (weighted averaging over a neighborhood of the above terms) involves an unweighted sum between two unit-less numbers and an energy term (in REU). It makes no sense physically to add them up, and this may result in suboptimal performance. Weights should be introduced here (e.g. by dividing by the standard deviation over the dataset of each term). Another option is to compute three distinct values (one averaging for each term) and calculate 6 features per residue.

5) The caption of Sup Fig1, writes: “Per-residue Rosetta Relative Energy Unit (REU) distributions. Since each conformation is associated with a unique relative free energy, we used REU to approximate the relative stability of each residue given the sum of its single-body and pairwise interactions and to account for conformation specificity.” How is “relative stability” computed? Is there a reference conformation? Please clarify and move to Methods.

Regarding the data:

1) the source of the epitope data is not provided. I did not have access to the Online Methods if there was any.

2) The definition used (distance cut-off criteria) is not specified.

3) my impression is that the epitope data is incomplete. For instance, for SARS-CoV-2 spike protein, the test set (https://github.com/mfsfischer/AxIEM/blob/main/testing_datasets/SARS2_epi.csv) contains 525 positive labels for 21123 negative ones. Given that there are 8 trimers here, this leaves 525/8*3 ~ 22 unique epitope residues per chain. I am certain that they are many more epitope residues, based on available crystal structures or a quick iedb search. Same for SARS-CoV-1 (15 epitopes). RSV and HIV datasets looked reasonable.

4) The definition of conformation-specific epitope L374 is unclear. What quantitative criteria are used to define structural similarity and/or antibody overlap with the antigen? Are the results shown for conformation-specific epitopes only, or for all epitopes?

5) There are hundreds of structures of fusion protein structures available on the pdb. What was the rationale for selecting these seven structures?

Manuscript presentation:

The manuscript features multiple 40/50+ word sentences that are very hard to parse/understand. Ex: L306-309.

L49: typo “techniques, ,”

L51-55: sentence unclear.

L65-67: sentence unclear

L161-164: sentence unclear. English; “despite that”

L203-207: The sentence and subsequent discussion cannot be understood. “the closed conformation … were labelled to be antigenic, while only the 2 or 3-RBD up … were labelled to be antigenic”.

L239-245. I could not understand neither the definition of FP_{FP <-> FN} and FN_{FP <-> FN}, nor the point that the authors were trying to make here.

L269 typo

L295: elicit

Eqn 3 L338: The left-hand term of the equation (NV_i) is a scalar, whereas the right-hand is a vector. A norm operation is missing?

Eqn 4 L 352: the indexing of the equation is not consistent with the text description. The inner sum should be over the conformation index, the outer sum over the neighbor index j. One running index for conformations k=1..n should be introduced.

L354: Terminology; ( Av(CP(ji)) – CP(ji)_n )^2 is not the variance of the CP value but the square deviation (the variance corresponds to average of the square deviation).

Reviewer #2: The authors describe a new algorithm, called AxIEM, that predicts antibody epitopes based on analysis of antigen structures and Bayes classification. As noted by the authors, such a method could be useful in prospective vaccine design efforts. While this method seems interesting, and it appears to compare well with the other methods that were tested (Discotope, Ellipro), the authors should provide readers with more information regarding the methods, including the training and testing regimen used to generate the results. Specific comments are noted below:

1. The authors should clearly state what was used for training and the withheld test sets for their results. It is not clear whether the AxIEM results shown in Table 1, Figure 2, and Figure 3 are from re-training of the model separately with distinct antigens and epitopes to apply to the test cases, or if there is a possible overlap between the training and the cases being shown in the results. One concern is that SARS-CoV and SARS-CoV-2 antigens could be present in both the training and test sets, and they have similar sequences and structures, as well as overlapping known epitopes in the RBD, etc. Additionally, there are two influenza HA antigens (HA stem from H3, and full H7 HA), which could also represent a possible redundancy in epitopes and structures, and it is not clear if they were separated during the training and test regimen. Details regarding the training and testing would be very helpful for readers to gauge the overall success of AxIEM and its comparison with Discotope and Ellipro, as nonredundancy between the train/test sets, or ideally the use of a withheld set, would be critical in gauging the predictive success of AxIEM on viral antigens not seen during its training.

2. The Methods section on “Definition of conformation-specific epitopes” requires more explanation and clarity. Regarding the experimentally validated epitopes used in this study, the authors note: “…database identifications have listed in Online Methods”. However, this information does not appear to be in the supplemental materials. Please provide more information where this information is located, or include it if it is not present.

3. This sentence from that Methods section (lines 374-378) requires clarification: “The definition of an epitope was further refined as any residue identified as a discontinuous epitope that also retained a structural similarity to the conformation of the antigen experimentally determined to interact with a specific antibody and that the binding orientation of the antibody did not occupy the same space, or intersect, with any part of the full-length viral protein in that conformation when aligned using PyMOL.” It is not clear what “specific antibody” the authors used for these comparisons; the set of these should be provided. Also, it is not clear what metric was used to define structural similarity (e.g. RMSD, within some cutoff distance). As the authors appear based on this description to utilize their own criteria to define epitopes, so they should list the final set of epitope residues for each antigen (used in their training and testing analysis) in their supplemental information.

4. The Introduction section needs more references to provide readers with context regarding several of the statements being made by the authors. For example:

- Lines 68-70: A citation should be added for the IEDB where it is noted.

- Lines 82-83: “Compared to other proteins with antigenic determinants within a viral quasispecies, fusion proteins are more frequently targeted by broadly neutralizing antibodies…”. One or more references should be added in support of this statement.

- Lines 84-85 (“…so-called ‘reverse vaccinology’.”). Please provide at least one reference regarding reverse vaccinology. One possibility is: Rappuoli et al. J Exp Med (2016) 213 (4): 469–481.

- Lines 88-90: “Recent advances in experimental design and cryogenic electron microscopy (cryo-EM) allow discovery of cryptic epitopes in ‘alternative’ conformations of viral fusion proteins.” A reference or multiple references providing examples for this would be very helpful here.

5. As noted by the authors (Line 410), the code and data for running this protocol are available on Github, which is great. However, the Github repository does not appear to contain any README file or information for interested readers on how to run the protocol. The authors should provide a README or documentation for this repository, in accordance with standard practices, to enable readers to understand and run the algorithm being described in this manuscript.

**Have the authors made all data and (if applicable) computational code underlying the findings in their manuscript fully available?**

Reviewer #1: Yes

Reviewer #2: **No: **The authors note "Online Methods" which should contain important information regarding their epitopes used in training and testing. As mentioned in my comment #2, this information could not be found.

PLOS authors have the option to publish the peer review history of their article (what does this mean?). If published, this will include your full peer review and any attached files.

Reviewer #1: No

Reviewer #2: No
---

## [Decision Letter · Decision Letter 1]

9 Nov 2022

Dear Dr. Fischer,

Thank you very much for submitting your manuscript "Computational epitope mapping of class I fusion proteins using low complexity supervised learning methods" for consideration at PLOS Computational Biology. As with all papers reviewed by the journal, your manuscript was reviewed by members of the editorial board and by several independent reviewers. The reviewers appreciated the attention to an important topic. Based on the reviews, we are likely to accept this manuscript for publication, providing that you modify the manuscript according to the review recommendations.

Sincerely,

Dina Schneidman

Software Editor

PLOS Computational Biology

Reviewer's Responses to Questions

**Comments to the Authors:**

Reviewer #1: The authors have thoroughly addressed my concerns and the manuscript is now substantially clearer and improved. I recommend publication as is.

Reviewer #2: The authors have addressed most of the questions and concerns from the initial review, including update of the Github and supplemental information. However, some concerns remain regarding the authors’ responses and revisions:

1. Response #16. The authors have indeed provided more detail for the “Definition of conformation-specific epitopes” methods. However, the test for van der Waals overlap seems very strict, particularly if it includes side chain atoms; one can imagine that normal or random movements of surface side chains, particularly large ones (e.g. Lys, Tyr) may result in this overlap being detected, versus bona fide antigen conformational changes. If these calculations with the 0.6 Å cutoff for any atom pair do indeed include side chains (versus just backbone atoms), the authors should briefly explain in the text of that section the possibility of potentially detecting side chain clashes during the calculations in some cases, and why this would not be a concern.

2. A minor comment related to the revised methods section is that “zero Å overlap” does not have a clear meaning. Revising the wording here e.g. to “no atomic overlap”, would be helpful for readers.

3. The description of the training and withheld sets is helpful, but the terminology “leave-out” is potentially confusing. It is not clear whether this refers to “leave-one-out” (which is a commonly used term), or more likely, a withheld or test set. The authors should change the “leave-out” wording to something more commonly used in machine learning studies, to avoid confusion, unless it indeed is commonly used and has sufficient precedent.

**Have the authors made all data and (if applicable) computational code underlying the findings in their manuscript fully available?**

Reviewer #1: Yes

Reviewer #2: Yes

PLOS authors have the option to publish the peer review history of their article (what does this mean?). If published, this will include your full peer review and any attached files.

Reviewer #1: No

Reviewer #2: No

Figure Files:

Data Requirements:

Reproducibility:

References:

---

## [Editor Report · Decision Letter 2]

24 Nov 2022

Dear Dr. Fischer,

We are pleased to inform you that your manuscript 'Computational epitope mapping of class I fusion proteins using low complexity supervised learning methods' has been provisionally accepted for publication in PLOS Computational Biology.

Best regards,

Dina Schneidman

Software Editor

PLOS Computational Biology

---

## [Editor Report · Acceptance letter]

2 Dec 2022

PCOMPBIOL-D-22-00778R2 

Computational epitope mapping of class I fusion proteins using low complexity supervised learning methods

Dear Dr Fischer,

I am pleased to inform you that your manuscript has been formally accepted for publication in PLOS Computational Biology. Your manuscript is now with our production department and you will be notified of the publication date in due course.

With kind regards,

Zsofi Zombor
